# Data From the MTurk Parent eHealth Preference Survey: Mental Health Symptoms, Psychosocial Correlates, Service Use, and Preferences

Journal of **open** psychology data

DATA PAPER

**CHARLIE RIOUX** ⓘD

**AVALINE KONKIN** ⓘD

**ANNA L. MACKINNON** ⓘD

**LIANNE TOMFOHR-MADSEN** ⓘD

**LESLIE E. ROOS** ⓘD

*Author affiliations can be found in the back matter of this article

]u[ ubiquity press

## ABSTRACT

The MTurk Parent eHealth Preference Survey aimed to examine parents' preferences for eHealth program features as well as their attitudes and preferences towards three programs: AbilitiCBT, Triple P Online, and BEAM. The data also cover sociodemographic factors, mental illness symptoms, social support, resilience, stressful experiences, medication use, and psychosocial service use. The data (n = 177; United States) were collected through an online cross-sectional self-report survey on MTurk. The primary and processed data are available OSF and have reuse potential for clinical research on parental mental health, methodological research on crowdsourcing for participant recruitment, and for use in statistics courses.

**CORRESPONDING AUTHOR:**

**Charlie Rioux**

Department of Interdisciplinary Human Sciences, Texas Tech University, US

charlie.rioux@ttu.edu

**KEYWORDS:**

depression; anxiety; parenting stress; treatment; mental health

**TO CITE THIS ARTICLE:**

Rioux, C., Konkin, A., MacKinnon, A. L., Tomfohr-Madsen, L., & Roos, L. E. (2025). Data From the MTurk Parent eHealth Preference Survey: Mental Health Symptoms, Psychosocial Correlates, Service Use, and Preferences. *Journal of Open Psychology Data,* 13: 2, pp. 1–11. DOI: https://doi.org/10.5334/jopd.127

# (1) BACKGROUND

Research and interest for internet- and mobile application-based health (eHealth) has been increasing exponentially in recent years (Rioux et al., 2022). While eHealth programs have been found to be a useful tool for promoting parental mental health and parenting skills, engagement is often found to be low, limiting the reach and impact of interventions (Florean et al., 2020). One factor that can be related to low engagement in current programs is limited knowledge on parents' preferences in eHealth programming. Indeed, research on in-person mental health services shows that programs that fit parent preferences have higher enrollment and adherence (Bannon & McKay, 2005; Nock & Kazdin, 2001) since they better align with the values, goals, and needs of parents accessing them (Cunningham et al., 2008, 2013; Hoagwood, 2005). However, despite increased interest and use of eHealth, knowledge about user preferences remains limited, especially those of parents seeking mental health and/or parenting support.

Preferences regarding specific program modalities and features can be examined to inform the details of program development. For example, understanding preferences regarding program length, targets, structure, and/or interactions with peers and clinicians, can directly inform the development, tailoring, and refinement of eHealth programs for parents. Beyond modalities and features, impressions and preferences based on existing programs can also be valuable, as this could inform how parents are guided by professionals in navigating current program offerings.

The collection of the data presented in this paper had the main aim of examining preferences between treatment modalities (e.g., preferences for eHealth vs. in-person options; for individual vs. group therapy; for interacting with parent peers; for program length, etc) as well as parents' impressions and preferences between three current programs: AbilitiCBT (myicbt.com), Positive Parenting Program (Triple P) Online (triplep-parenting.com), and Building Emotional Awareness and Mental Health (BEAM; thebeamprogram.com). AbilitiCBT is focused on mental health and users go through 10 weekly self-paced online modules that teach techniques to help individuals understand and cope with mental illness symptoms based on the principles of Cognitive Behavioral Therapy (CBT) while a therapist monitors progress. Triple P Online is focused on parenting and users work through eight online modules at their own pace (average completion time is 11 weeks), with modules featuring video clips, interactive exercises, and a workbook on how to improve the family environment and prevent and reduce behavioral problems in children (Baumel & Faber, 2018). BEAM is focused on both mental health and parenting and users go through a 10-week program with weekly group therapy sessions,

educational videos on mental health and parenting skills, a private online forum to connect with other parents, and optional work sheets (MacKinnon et al., 2022). While the data collection's main objective was to understand these preferences, many potential covariates of preferences and mental health were also collected, making the dataset of potential use for future research and secondary analyses.

# (2) METHODS

## 2.1 STUDY DESIGN

Data were collected for this project using a cross-sectional online survey. Participants were recruited through Amazon Mechanical Turk (MTurk) and provided with a link to access an online survey hosted on Qualtrics. Prior to beginning the study, all individuals were shown a consent form that provided an overview of the study. If individuals chose not to consent, the survey would shut down. Those who chose to consent continued to the eligibility screener. To be eligible to participate, participants needed to: (1) be comfortable understanding, reading, and speaking English; (2) have at least one child 0–5 years of age; and (3) be a resident of either Canada or the United States. If an individual did not meet all eligibility requirements, they were not given access to continue with the survey. Eligible participants answered sociodemographic questions, followed by questions on their general treatment and eHealth preferences and descriptions and questions on AbilitiCBT, BEAM, and Triple P Online. This was followed by questionnaires on parenting stress, anxiety, depression, social support, medication/service use, and stressful experiences. Participants were also provided with a list of resources for mental health concerns and family stress at the end of the survey.

## 2.2 TIME OF DATA COLLECTION

Data were collected from March 3 to March 14, 2022.

## 2.3 LOCATION OF DATA COLLECTION

While participation was open to residents of both Canada and the United States, only individuals from the United States participated. The geographical distribution of the sample can be found in Figure 1; participants were from 37 states.

## 2.4 SAMPLING, SAMPLE AND DATA COLLECTION

Participants were recruited through Amazon MTurk. MTurk is an online labor market that can be used as a crowdsourcing tool for research. Requesters post advertisements for short-term Human Intelligence Tasks (HITs) that workers complete for compensation. The worker is paid upon completion of the task and

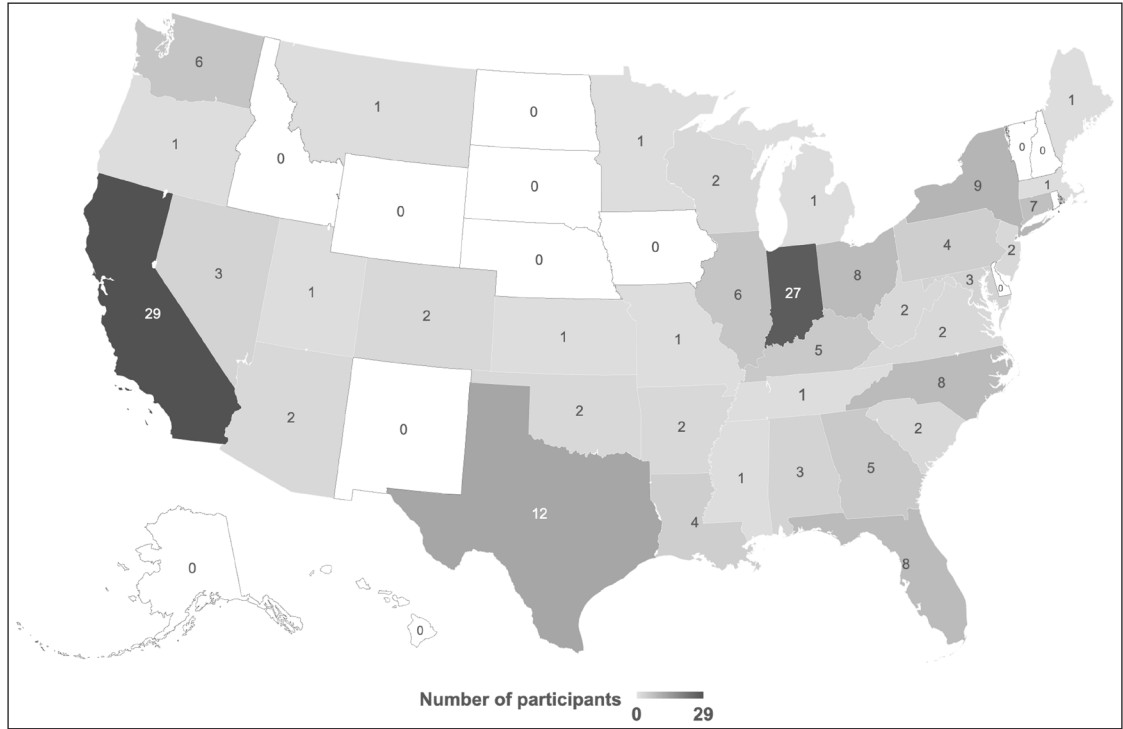

**Figure 1** Geographical distribution of study participants. Adapted from Rioux, Konkin et al (2024), published under Creative Commons Attribution 4.0 International License.

payment is handled by Amazon with a commission rate. Prior research has found MTurk to be an effective and efficient online parent recruitment strategy (Dworkin et al., 2016).

The MTurk HIT was restricted to MTurk workers with a Parenthood Status premium qualification in the United States or Canada (but only parents from the United States participated). Parents were eligible if they were comfortable understanding, reading, and speaking English, and had at least one child between 0–5 years of age. Once the survey had been completed, participants were provided with a random ID which they were instructed to enter in the applicable space on MTurk in order to receive their compensation of $6USD.

Unreliable data (from inattention, careless responding, and web robots) were identified through attention checks, mismatches between country and province/state (e.g., selects US and a Canadian province for their current residence), and completion times under 10 minutes, which were considered impossible based on survey testing. Our HIT for 300 responses yielded 335 completed surveys from eligible participants, of which 47.2% were deemed unreliable based on the criteria listed above. Thus, the final sample consists of 177 parents. The sample demographics can be found in Table 1.

## 2.5 MATERIALS/SURVEY INSTRUMENTS

Unless indicated otherwise, questionnaires are listed below in the order of the online survey as seen by participants. When questionnaires pertained to a child or the relation of parents with their child, participants were asked to think about their oldest child between 0–5 years of age.

### 2.5.1 Sociodemographic questionnaire

The survey began with questions on demographic factors. The following variables were assessed: state currently lived in, age, gender identity, sex assigned at birth, ethnic background, highest level of education attained, marital status, daily activities and responsibilities, household income, and number of children. Four questions were asked regarding participants' oldest child aged between 0–5 years: month and year of birth, relation to the child, how often the child resides in the participant's household, and if the participant views themselves as the primary caregiver for the child.

### 2.5.2 eHealth preferences questionnaire

The eHealth preferences questionnaire was developed by the research team to meet the main objectives of the original study. In this section, participants first answered questions about treatment modality preferences. In the first three questions, participants ranked their preferences for types of support. The first question asked if they were seeking professional support, would they prefer receiving mental health support only, parenting support only, or both. These options were ranked from 1 = most preferred to 3 = least preferred. The second question asked the type of help they would most prefer receiving if they were seeking mental health support (prescription medication, in-person individual counselling, in-person support group, or app-based/internet-based program), ranked from 1 = most preferred to 4 = least preferred. The third question asked the type of help they would most prefer receiving if they were seeking parenting support (in-person individual counselling, in-person support group, or app-based/internet-based program),

| VARIABLE | MEAN (SD) | n (%) |
|---|---|---|
| Age | 32.6 (8.2) | |
| Sex assigned at birth | | |
| Female | | 81 (45.8%) |
| Male | | 95 (53.7%) |
| Gender | | |
| Cisgender woman | | 79 (44.6%) |
| Cisgender man | | 93 (52.5%) |
| Transgender woman, transgender man, or non-binary | | 5 (2.8%) |
| Ethnicity | | |
| White American | | 122 (68.9%) |
| White European | | 28 (15.8%) |
| White African | | 10 (5.6%) |
| Black African American | | 6 (3.4%) |
| Black African | | 6 (3.4%) |
| Latin American | | 11 (6.2%) |
| East Asian, Southeast Asian, Indo-Caribbean, or Middle Eastern | | 8 (4.5%) |
| Education | | |
| High school diploma or less | | 12 (6.7%) |
| Technical diploma, associate degree, or undergraduate certificate | | 12 (6.8%) |
| Bachelor's degree | | 101 (57.1%) |
| Master's degree | | 52 (29.4%) |
| Marital status | | |
| Single, never married, separated | | 10 (5.7%) |
| Married or domestic partnership | | 166 (93.8%) |
| Pretax household income | | |
| 20 000$–39 999$ | | 20 (11.3%) |
| 40 000$–69 999$ | | 70 (39.5%) |
| 70 000$–99 999$ | | 53 (29.9%) |
| 100 000$–124 999$ | | 16 (9.0%) |
| 125 000$–149 999$ | | 10 (5.6%) |
| 150 000$+ | | 8 (4.5%) |
| Number of children | 1.5 (0.6) | |
| **Characteristics of oldest child between 0–5 years old** | | |
| Age | 3.3 (1.3) | |
| Relation to the participant | | |
| Biological child | | 170 (96.0%) |
| Adoptive or foster child | | 7 (4.0%) |
| Frequency of living with the participant | | |
| Full-time | | 167 (94.4%) |
| Part-time | | 7 (4.0%) |
| Participant views themselves as the primary caregiver | | |
| Yes | | 149 (84.2%) |
| No | | 5 (2.8%) |
| Shared equally with another parent | | 23 (13.0%) |

**Table 1** Sample demographics.

*Note.* Table reproduced from Rioux, Konkin et al. (2024), published under Creative Commons Attribution 4.0 International License.

ranked from 1 = most preferred to 3 = least preferred. For all three questions, additional variables were computed representing the most preferred modality (i.e., based on the modality ranked 1).

After completion of the previous section, participants answered an additional nine questions where they were provided features that virtual mental health and parenting support platforms may use and asked to rate how likely they would be to use a virtual mental health and parenting support platform that included the feature on a Likert scale from 1 (very unlikely) to 5 (very likely). These features include (1) weekly access to a clinically-trained coach, (2) occasional individual sessions with a clinically-trained coach, (3) weekly group sessions with a clinically-trained coach, (4) information provided through videos, (4) information provided through reading materials, (6) weekly activities such as worksheets, practices exercises, (7) ability to track own symptoms/progress, (8) having a clinically-trained coach monitor their symptoms/progress, and (9) weekly opportunities to connect with other parents in the program. Participants were then asked what length of a virtual mental health and parenting support program would best fit their life on a visual analog scale going from 6 to 20 weeks.

The next set of questions asked about how and with whom participants would like to connect when using a virtual mental health and parenting support platform. The first questions asked if they would prefer connecting with other parents in the program anonymously, non-anonymously (using personally identifying details such as their name), or if they would prefer to not have contact with other parents in the program. Next, for forums and videoconferencing separately, participants were asked if they would be comfortable connecting with (1) mothers only, father only, or all parents; and (2) biological, adoptive, foster, and/or step-parents.

The last section on eHealth preferences focused on the comparison of AbilitiCBT, Triple P Online, and BEAM. Participants were then given nameless descriptions of each of the three programs. These descriptions can be found in the study's OSF project at https://doi.org/10.17605/OSF.IO/J7D2Q and in Rioux, Konkin et al., 2024. The descriptions were drafted and reviewed by three clinical psychologists with expertise in mental health and parenting interventions. Consistent with previous research (e.g., Cameron et al., 2017; Goodman et al., 2013), each description included six pieces of information: What is the approach? How does the approach conceptualize mental health/depression or parenting stress/problems? How does the approach work? What will I do in this approach? How much time is involved? What are the risks?

After reading each description, participants ranked on a visual analog scale ranging from 0 to 100 how likely they would be to enroll in the program and how likely they would be to complete the program. This

was followed by the credibility and personal reactions to rationales scales, which were adapted from the original scale developed for depression research (Addis & Carpenter, 1999), where "parenting stress, depression, and/or anxiety" was used in place of "depression." The Credibility Scale (Addis & Carpenter, 1999) assessed the extent to which participants perceived each program to be logical, complete, scientific, and helpful in other areas of life, and how likely they would be to recommend the approach. The Credibility Scale consists of seven items rated on a Likert scale from 1 (not at all) to 7 (extremely). Mean scores were computed, with higher scores indicating greater credibility. Reliability was good in this sample (Cronbach's α AbilitiCBT = .88; BEAM = .89; Triple P = .92). The Personal Reactions to Rationales Scale (Addis & Carpenter, 1999) assessed participants' perceptions of each program in terms of (1) how helpful they expect the program would be for them, (2) the extent to which the approach would help them to understand the causes of parenting stress, anxiety and/or depression, (3) the extent to which the approach would help them learn to cope with parenting stress, anxiety and/or depression, (4) how likely they would be to choose the approach, and (5) how effective they think the approach would be in treating parenting stress, anxiety and/or depression. Items were answered on a Likert scale from 1 (not at all) to 7 (extremely). Mean scores were computed, with higher scores indicating more positive personal reactions. Reliability was good in this sample (Cronbach's α AbilitiCBT = .86; BEAM = .90; Triple P = .89).

After finishing these scales, participants were asked to indicate which program they would be most likely to want to take part in if they were given the choice between the three programs.

### 2.5.3 Parenting stress
Parenting stress was measured using the Parenting Stress Index-Short Form (PSI-SF; Abidin, 2012). The PSI-SF consists of 36 items rated on a Likert scale from 1 (Strongly disagree) to 5 (Strongly agree) assessing parent stress and interactional style in three domains: how parents feel in their role, how satisfied they are in the relationship with their child, and how difficult they perceive their child to be. Sum scores for the total scale were computed. Reliability was good in this sample (Cronbach's α = .97). Dichotomous scores were then computed based on the scale's established cut-off scores (Abidin, 2012) for mild parenting stress (36–89) and clinically concerning parenting stress (90+).

### 2.5.4 Anxiety and Depression
General anxiety symptoms were measured using the Generalized Anxiety Disorder 7-Item Scale (GAD-7; Spitzer et al., 2006). The GAD-7 assesses general anxiety symptoms over the last two weeks, rated on a Likert scale from 0 (not at all) to 3 (nearly every day).

Sum scores were computed. Reliability was good in this sample (Cronbach' α = .90). Dichotomous scores were also computed based on the scale's established cut-off scores (Spitzer et al., 2006) for mild anxiety symptoms (0–9) and moderate-high anxiety symptoms (10+).

Depression symptoms were measured using the Patient Health Questionnaire-9 item (PHQ-9; Kroenke et al., 2001). The PHQ-9 assesses depression symptoms over the last two weeks, rated on a Likert scale from 0 (not at all) to 3 (nearly every day). Sum scores were computed. Reliability was good in this sample (Cronbach's α = .88). Dichotomous scores were also computed based on the scale's established cut-off scores (Kroenke et al., 2001) for mild depression symptoms (0–9) and moderate-high depression symptoms (10+).

Continuous scores of depression and anxiety were highly correlated (r = .88, p < .001), thus mental health symptom scores combining anxiety and depression were also computed. To make the mental health symptom continuous score, the PHQ-9 and GAD-7 were combined by standardizing them (since they are on different scales) and then averaging them. A categorical dummy score was also computed based on the clinical cut-offs listed above, with 0 = mild mental health symptoms, and 1 = clinically concerning mental health symptoms (depression and/or anxiety is moderate-high).

### 2.5.5 Social support
Perceived adequacy of social support from family, friends, and significant others was measured using the Multidimensional Scale of Perceived Social Support (Zimet et al., 1988, 1990). This scale consists of 12 items answered on a Likert scale from 1 (very strongly disagree) to 7 (very strongly agree). Mean scores were computed with higher scores representing higher perceived levels of social support. Reliability was excellent in this sample (Cronbach's α = .92).

One separate question at the end of the stressful experiences questionnaires (see section 2.5.7) asked "do you have someone with whom you can share your deepest thoughts and feelings?", with Yes/No answer options.

### 2.5.6 Medication and service use
This section first asked about medication use. Participants were asked if they were currently prescribed any psychiatric medications (yes/no). If they answered yes, they were asked to check all that apply among antidepressants, antianxiety/antipanic medications, stimulants, mood stabilizers, and other (with an open-text field if they checked other). The rest of this section asked about a number of mental health services that they may have accessed. For each service, they were asked to rate how often they utilized these services over the last month (did not access the service; or time accessed in days, average time per use

in minutes). They were then asked to rate how satisfied they were with the service from 1 (minimally) to 5 (very much). The services assessed included (1) virtual or in-person individual counselling or therapy; (2) virtual or in-person group counselling; (3) instant messaging mental health services; (4) mental health crisis line; (5) seeking mental health information online; (6) faith-based counselling services with religious leaders (7) app-based or online mental health program; (8) other, with option to specify.

### 2.5.7 Stressful experiences questionnaires

Exposure to recent stressful experiences was measured using the Recent Stressful Experiences checklist (Roos et al., 2021), which was created based on recommendations from the JBP Research Network on Toxic Stress (Center on the Developing Child, 2021). Using 10 questions answered with "Yes" or "No," this questionnaire assesses the presence or absence of stressors in the last twelve months (i.e., life threatening illness or accidental injury to you or someone close to you, death of someone close to you, moved to a different home or apartment, family violence or abuse to you or someone close to you, had to take care of a seriously ill or disabled family member, started back at school, separated or divorced with a spouse or romantic partner, you or someone in your home lost a job or tried to get a job and failed, government agency funds were cut off for you or someone in your home, anything else bad happened to you or someone close to you). These items were summed to obtain a cumulative recent stressful experiences score ranging from 0 to 10. Reliability was acceptable in this sample (Cronbach's $\alpha$ = .71).

Resilience was measured with the Response to Stressful Experiences Scale-4 items (RSES-4; De La Rosa, Webb-Murphy & Johnston, 2016). This scale measures resilient coping tendencies, or coping behaviours that people may engage in when under pressure using the prompt "During and after life's most stressful events, I tend to…" with four items (e.g., find a way to do what's necessary to carry on) rated from 1 (not at all like me) to 5 (exactly like me). A multi-sample validation study found a Cronbach's $\alpha$ ranging from 0.73 to 0.88 (De La Rosa, Webb-Murphy & Johnston, 2016). The scale score was not computed in the provided dataset, but all items are available.

### 2.6 QUALITY CONTROL

Best-practice MTurk recommendations (Aguinis et al., 2021) were employed, including a captcha verification and attention checks. Two attention checks were included using instructed response items (Gummer, Roßmann & Silber, 2018). Specifically, "In this question, we ask you to choose six (6) on the answer scale to show that you have read this sentence" was included within the credibility scale of the BEAM program (section 2.5.2) and "Please

select Several days" was included between the fourth and fifth questions of the GAD –7 (section 2.5.4). Unreliable data (from inattention, careless responding, and web robots) were identified through these attention checks as well as mismatches between country and province/state (e.g., selects United States and a Canadian province for their current residence), and completion times under 10 minutes, which were considered impossible based on survey testing (see section 2.4).

The reliability of questionnaires/scales within the study sample was checked using Cronbach's alphas (see section 2.5). During scale computations, scales available in the processed data were checked for univariate outliers by examining the combination of standardized values (Z-scores) exceeding ± 3.29 and being out of the distribution of other scores on the variable (Tabachnick & Fidell, 2019). Identified univariate outliers' raw score was recoded to be a unit larger or smaller than the next most extreme score (Tabachnick & Fidell, 2019). This process is included in the provided syntax files (sections 3.2, 3.3).

### 2.7 DATA ANONYMISATION AND ETHICAL ISSUES

Ethics approval was granted by the University of Manitoba Psychology/Sociology Research Ethics Board (HE2021–0090). All participants provided electronic informed consent prior to answering study questionnaires by reviewing and agreeing to the project's consent form on Qualtrics. Consent included de-identified data being made available on public data platforms such as open science framework (OSF). The consent form is provided in the study materials of the OSF project found at https://doi.org/10.17605/OSF.IO/J7D2Q (see Table 2). All participants were given a random ID for the dataset to protect their identity.

A few variables had to be removed from the public dataset to anonymise the data due to their potential for participant identification through cross tabulation. This includes location data and variables with low counts (under n = 10). All automatically collected Qualtrics (e.g., geolocation) data except survey duration in seconds and recorded date was removed. Sex assigned at birth was removed due to potential identification of one intersex participant and participant gender identity was removed due to the potential identification of gender-diverse participants. A computed variable representing cisgender mothers and fathers only remains in the clean dataset. The ethnicity original variable was removed due to low category counts. An ethnicity variable with less categories was computed specifically for the shared dataset and can be found in the processed dataset. Only the White/Black categories could be maintained as other ethnicities had counts under n = 10. Two variables are provided: (1) Participants identifying as White Americans only (i.e.,

| FOLDER NAMES | FILE NAMES | DESCRIPTION | DATA TYPE |
|---|---|---|---|
| Data and analyses | ParentPref_OriginalData.sav ParentPref_OriginalData.csv | Primary data file, excluding identifying information (section 2.7) | Primary data |
| | ParentPref_OriginalData_Labels.pdf | Variable label information for ParentPref_OriginalData.csv | Codebook |
| | ParentPref_CleanData.sav ParentPref_CleanData.csv | Data following sample processing (section 2.4), scale computations (section 2.5 and 2.7), and Quality control (section 2.6) | Processed data |
| | ParentPref_CleanData_Labels.pdf | Variable label information for ParentPref_CleanData.csv | Codebook |
| | CleanData_Syntax.sps | SPSS syntax to make processed data from primary data. | Code |
| | Clean Data_Syntax_Annotated.pdf | Annotated PDF version of CleanData_Syntax.sps | Code |
| | CleanData_Output.spv | Output from CleanData_Syntax.sps | Output |
| Study materials | ConsentForm.docx ConsentForm.pdf | Consent form completed by participants. | Materials |
| | MTurk_HIT_Posting.pdf | Advertisement posted on Amazon MTurk to recruit participants. | Materials |
| | Questionnaires.docx Questionnaires.pdf | Study questionnaires | Materials |

**Table 2** OSF project file descriptions for primary and processed data.

Note. Files listed are for dataset use. Additional files in the OSF project are from analyses conducted for the peer-reviewed publications listed in section 2.8.

not multiracial) and (2) Participants identifying as White or Black only (other participants have missing data). The original income variable (Table 1) was removed due to low category counts. It was recoded for the shared dataset, merging the 150000$+ category with the previous category, and can be found in the processed dataset. Variables identifying relation to the child, how often the child resides in participant's household, and if participant views themselves as primary caregiver were removed due to low category counts. Exact child month and day of birth were also removed as they are potentially identifying. The child's age was computed and is available in the processed dataset.

## 2.8 EXISTING USE OF DATA

As of summer 2024, data obtained from this project has been used to contribute to the following conference abstract:

Rioux, C., Konkin, A., MacKinnon, A. L., Cameron, E. E., Tomfohr-Madsen, L. M., Watts, D. & Roos, L. E. (2022). *Parent Preferences for Peer Connection in eHealth Programs.* Iproceedings. *8*(1). e39278 https://doi.org/10.2196/39278

Additionally, the data were used for the two following peer-reviewed papers:

Rioux, C., Childers-Rockey, Z. A., Konkin, A., Cameron, E. E., Tomfohr-Madsen, L. M., MacKinnon, A. L., Watts, D., Murray, J., Pharazyn, A. & Roos, L. E. (2024). Parent Preferences for Peer Connection in Virtual Mental Health and Parenting Support. *Journal of Technology in Behavioral Science.* https://doi.org/10.1007/s41347-024-00408-8

Rioux, C., Konkin, A., Tomfohr-Madsen, L., MacKinnon, A. L., Cameron, E. E, Watts, D., Xie, E. B. & Roos, L. E. (2024). Parent eHealth preferences: Perceived credibility and personal reactions to AbilitiCBT, BEAM, and Triple P Online. *Collabra: Psychology. 10*(1); 121252. https://doi.org/10.1525/collabra.121252

## (3) DATASET DESCRIPTION AND ACCESS

### 3.1 REPOSITORY LOCATION

https://doi.org/10.17605/OSF.IO/J7D2Q

### 3.2 OBJECT/FILE NAME

See Table 2.

### 3.3 DATA TYPE

The data and detailed versions of the study materials are available. See Table 2.

### 3.4 FORMAT NAMES AND VERSIONS

All data are available in SPSS and comma separated values (CSV) formats.

### 3.5 LANGUAGE

English

### 3.6 LICENSE
CC-By Attribution 4.0 International

### 3.7 LIMITS TO SHARING
There is no limit to sharing.

### 3.8 PUBLICATION DATE
17/05/2024

### 3.9 FAIR DATA/CODEBOOK
Descriptive metadata were included in the OSF repository, including project description, subjects, and tags. A digital object identifier (DOI) was assigned to the project to support permanent identification. The use of a reputable and widely used repository (OSF) supports accessibility. Downloading the data on OSF does not require identification or authorization procedures, in turn supporting accessibility. The OSF project includes README files to aid in navigating the project folders. SPSS data files were saved and uploaded in CSV format for accessibility. SPSS data files (Table 2) include variable information in the variable labels tab. The variable information was also saved in PDF format for both the primary data and the processed data (Table 2) as data codebooks for use with the CSV files. The study materials, including the consent form, MTurk HIT posting (i.e., study advertisement), and questionnaires are also provided in the OSF project (Table 2).

## (4) REUSE POTENTIAL

The present data are of potential interest to clinical, health, and developmental psychology researchers, other mental health professionals, and statistics educators.

### 4.1 KEY STRENGTHS AND LIMITATIONS OF THE DATA
This dataset has some limitations in terms of the population it generalizes to. Results are limited to predominately white American/European and highly educated parents. This limits generalizability of eHealth program preferences as it has been found that education and racial-ethnic background are related to treatment preferences, with less educated parents preferring briefer intervention modules with concise information (Broomfield et al., 2022) and some communities preferring culturally-informed interventions (Toombs et al., 2018; Zhou et al., 2018). Furthermore, MTurk worker parents have been found to have high levels of distress and to be closer to an at-risk or clinical sample than a community sample (Jensen-Doss et al., 2022), and the sample of the current data having high levels of clinically concerning depression/anxiety (70.1%) and parenting stress (74.6%). An important strength of this study for research purposes is the inclusion of all genders, as most

research on parents has focused on "primary parents," most often mothers (Nomaguchi & Milkie, 2020).

### 4.2 RESEARCH ON PARENT MENTAL HEALTH
There is a wide range of reuse potential for secondary analyses. While the main objectives and published results focused on eHealth preferences, this data can be used for research on parent mental health (depression, anxiety, parenting stress) or mental health medication/ psychosocial service use. A range of correlates of mental health and medication/service use can be examined through the present data, including sociodemographic factors, family/significant other/friend social support, stressful experiences, and resilience.

The dataset also has some potential for examining diathesis-stress or environmental sensitivity (Pluess, 2015), or how individual temperament moderates the influence of the environment on mental health, as the RSES-4 (resilience scale) could be used as a moderating factor of environmental factors on mental health.

The data can also be used for individual participant-level meta-analyses (Burker, Ensor & Riley, 2017; Sutton, Kendrick & Coupland, 2008) related to parent mental health or parent treatment preferences, as well as harmonized with other datasets for integrative data analyses (Hussong, Curran & Bauer, 2013) related to parent mental health.

### 4.3 RESEARCH ON CROWDSOURCING FOR PARTICIPANT RECRUITMENT
Since the present data were collected through Amazon MTurk, it could also be used for methodological research on crowdsourcing platforms for participant recruitment. This includes research examining MTurk participant demographics and psychological characteristics or comparing the characteristics of samples recruited through different methods or platforms. The data could also be used for research testing new data processing and cleaning methods (Chu et al., 2016) for crowdsourced data.

### 4.4 PSYCHOLOGY STATISTICS COURSES
The present data can be used for advanced undergraduate or first-year graduate statistics courses teaching descriptive statistics, t-tests, chi-square tests, ANOVA models, correlation, regression and/or basic psychometrics. Considering data-driven courses and the emphasis of student practice with data are often recommended and found to increase student motivation (Hall, 2011; Smith, 1998; Wiberg, 2009), the current data could be used to fulfil those objectives. Practice with real-life data can enhance student interest and engagement (Idris, 2018), thus this data could be used for clinical psychology students or for general psychology courses along with other datasets to match the varied interests of the students. Since both the primary and processed

data are provided, the present dataset can be used for classes that begin with data processing/cleaning and classes that do not cover this part of data analysis.

## FUNDING INFORMATION

This data collection was funded by a grant from Research Manitoba to LT-M and LER.

## COMPETING INTERESTS

The authors have no competing interests to declare.

## AUTHOR CONTRIBUTIONS

Conceptualization: CR, ALM, LT-M, LR
Methodology: All authors
Software: CR, AK
Formal analysis: CR
Investigation: CR, AK
Resources: LR
Data curation: CR
Writing – Original draft: CR, AK
Writing – Review and editing: All authors
Visualization: CR
Project administration: CR, LR
Funding acquisition: LT-M, LR

## AUTHOR AFFILIATIONS

**Charlie Rioux** orcid.org/0000-0002-7943-0710
Department of Interdisciplinary Human Sciences, Texas Tech University, US

**Avaline Konkin** orcid.org/0000-0003-3107-2261
Department of Psychology and Neuroscience, Dalhousie University, CA

**Anna L. MacKinnon** orcid.org/0000-0002-1705-2048
Department of Psychiatry and Addictology, University of Montreal, CA; CHU Ste-Justine Research Centre, CA

**Lianne Tomfohr-Madsen** orcid.org/0000-0002-0860-5392
Department of Educational and Counselling Psychology, and Special Education, University of British Columbia, CA

**Leslie E. Roos** orcid.org/0000-0001-7083-4017
Department of Psychology, University of Manitoba, CA; Children's Hospital Research Institute of Manitoba, Children's Hospital Foundation of Manitoba, CA

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

## PEER REVIEW COMMENTS

*Journal of Open Psychology Data* has blind peer review, which is unblinded upon article acceptance. The editorial history of this article can be downloaded here:

- **PR File 1.** Peer Review History. DOI: https://doi.org/10.5334/jopd.127.pr1

**TO CITE THIS ARTICLE:**
Rioux, C., Konkin, A., MacKinnon, A. L., Tomfohr-Madsen, L., & Roos, L. E. (2025). Data From the MTurk Parent eHealth Preference Survey: Mental Health Symptoms, Psychosocial Correlates, Service Use, and Preferences. *Journal of Open Psychology Data,* 13: 2, pp. 1–11. DOI: https://doi.org/10.5334/jopd.127

**Submitted:** 15 October 2024      **Accepted:** 07 April 2025      **Published:** 14 April 2025

