## [Peer Review History. · Journal of Open Psychology Data]

Response to reviewer comments

We thank the reviewers for their positive comments and suggestions for improvement, which we believe improved the manuscript. Please find answers to each comment below.

REVIEWER 1

Major comments:

Comment 1: While the SPSS syntax file is extremely helpful for researchers familiar with SPSS, not everyone uses this software. Could the authors consider providing a plain-text summary of the cleaning steps, such as in a PDF or Markdown file, to make the process more accessible? Highlighting key recoding steps, labeling changes, and the reasoning behind merging or discarding categories would allow researchers using tools like R, Python, or Stata to replicate the workflow. This would support open science and make the data more widely usable.

Response: We added an annotated PDF file of the SPSS syntax as recommended by the reviewer. The README file (OSF) and Table 2 in the manuscript were updated accordingly.

Comment 2: The ParentPref_CleanData_Labels.pdf is a valuable resource, offering clear mappings between variables and their labels (e.g., "1" = "PROVINCES"). Including a full list of all possible codes, even those not present in the final dataset, and clarifying whether certain codes represent missing or unused categories, could make it even more helpful. A concise table in the README file summarizing frequently used variables might also be useful for first-time users.

Response: All labels are provided in the label file. The variable "State" (which has a first value of 1=PROVINCES), as noticed by the reviewer, seems incomplete (e.g., there is no 2 or 3) but this is due to how it was coded in Qualtrics (the original values skipped numbers). We in the README file that "all original survey variable values (present or not in the dataset) are included". Based on this comment, we also added in the README file that missing values are blank cells in the dataset. Finally, we added the labels of the most frequently used variables in the README file.

Comment 3: The manuscript mentions several standardized instruments (e.g., PHQ-9, GAD-7) but doesn't include reliability estimates (like Cronbach's alpha). Adding these details could help readers understand how well these measures performed in this sample and enhance the dataset's value for comparisons or meta-analyses. Perhaps this could be included in the methods or supplementary materials.

Response: Cronbach's alphas are provided when questionnaires are described in section 2.5.

Comment 4: The paper mentions that participants were predominantly White American/European and highly educated, which limits generalizability. Could the authors reflect on how this sample's demographic composition—such as its racial/ethnic

homogeneity and higher education levels—might influence the findings? For instance, would preferences for eHealth programs differ for parents with lower educational attainment or from minority racial/ethnic groups? A brief discussion of these points could help contextualize the findings and guide future research.

Response: We added a brief discussion of how the sample being predominantly white and educated relates to limited generalization of eHealth preferences.

Minor comments:

Comment 5: In the introduction, the sentence begins with:

“One factors that can be related to low engagement in current programs...”

This should be revised to:

“One factor that can be related to low engagement in current programs...”

Response: The typo was corrected.

Comment 6: The manuscript alternates between using "data was" and "data were." While modern usage sometimes accepts "data" as singular, scientific writing often treats it as plural. Could the authors ensure consistency throughout the text for clarity?

Response: All instances of “data was” were changed to “data were”.

Comment 7: The inclusion criteria specify that participants were “comfortable understanding, reading, and speaking English and had at least one child...” Adding a comma for clarity—“participants were comfortable understanding, reading, and speaking English, and had at least one child...”—might make the sentence easier to follow.

Response: A comma was added between “English” and “and”.

REVIEWER 2

Methods

The methods section of the paper is already quite detailed so that the reader can understand how the dataset was created. Only in a few places could additional information be added to improve understanding.

Comment 1: The Background section contains detailed information on the eHealth programs examined. In particular, the time frame of BEAM is described in detail. In the case of AbilitiCBT, it can be seen that there are 10 online modules, but here, as with the 8 modules of Triple P Online, the time aspect is not directly apparent. Additional

information could give the reader a better insight into the treatments.

Response: The time of completion of AbilitiCBT and Triple P were added to the introduction.

Comment 2: In the Study design section, a brief addition of information about what exactly Amazon Mechanical Turk (MTurk) is, could help international recipients to better understand the recruitment method.

Response: A brief description of MTurk was added at the beginning of section 2.4.

Comment 3: Figure 1 shows the graphical distribution of the participants. However, the legend of the graph could be misleading, as the title 'Number of parents' raises the question of whether it refers to surveyed pairs of parents or individuals.

Response: "parents" was changed to "participants" in the figure legend.

Comment 4: In Table 1, which shows the demographic distribution, indentations of the subcategories (as in the study from which the table was reproduced) could make the table appear visually clearer.

Response: The formatting of the table was corrected.

Comment 5: In Section 2.7, the sentence 'Consent included de-identified data being made be made available on public data platforms such as open science framework (OSF).' contains a doubling.

Response: The sentence was corrected.

Dataset

Comment 6: The document contains all the important information about the dataset or indicates where it can be accessed. The data is deposited under an open license that allows unrestricted access. *A brief marginal note outside of Table 2 to the effect that the detailed version of the study materials can also be viewed under the link could possibly be additionally informative for the reader.* The data is well stored under the link provided and can even be viewed in part without downloading. Another strength is the indication of three studies that have already worked with the data.

Response: We added that the detailed version of the study materials can also be viewed in the repository under section 3.3.

Reuse section

The reuse section provides concrete and useful suggestions for reuse of the data. Limitations of the data are pointed out and their strengths are highlighted. Possible future research approaches are outlined.

Overall, the paper is very informative, clearly structured and provides the reader with all the information necessary to understand and use the dataset. Only minor changes or

additions (as mentioned above) can provide even better access to the data set for readers who are not fully informed on the subject.